# Ceftolozane/Tazobactam In-Vitro Activity against Clinical Isolates from Complicated Intra-Abdominal Infection Patients in Three Indonesian Referral Hospitals

**DOI:** 10.3390/antibiotics12010052

**Published:** 2022-12-28

**Authors:** Anis Karuniawati, Merry Ambarwulan, Selvi Nafisa Shahab, Yefta Moenadjat, Toar J. M. Lalisang, Neneng Dewi Kurniati, Vicky Sumarki Budipramana, Tomy Lesmana, Iva Puspitasari, Erik Prabowo, Dwi Putranti Chitra Sasmitasari, Dian Oktavianti Putri, Amrilmaen Badawi

**Affiliations:** 1Department of Microbiology, Faculty of Medicine, Universitas Indonesia—Cipto Mangunkusumo Hospital, Jakarta 10430, Indonesia; 2Department of Surgery, Faculty of Medicine, Universitas Indonesia—Cipto Mangunkusumo Hospital, Jakarta 10430, Indonesia; 3Department of Medical Microbiology, Faculty of Medicine, Airlangga University, Dr. Soetomo General Academic Hospital, Surabaya 60286, Indonesia; 4Division of Digestive Surgery, Department of Surgery, Faculty of Medicine, Airlangga University, Dr. Soetomo General Academic Hospital, Surabaya 60286, Indonesia; 5Dr. Kariadi General Hospital, Semarang 50244, Indonesia; 6Faculty of Medicine, Diponegoro University, Semarang 502775, Indonesia; 7Global Medical Scientific Affair, Merck Sharp & Dohme (MSD), Jakarta 10220, Indonesia

**Keywords:** ceftolozane/tazobactam, complicated intrabdominal infection, *Enterobacterales*, *Pseudomonas aeruginosa*, Indonesia, antimicrobial susceptibility

## Abstract

Complicated intra-abdominal infections (cIAIs) lead to high morbidity and mortality, especially if poorly managed. However, Indonesia’s microbial pattern and susceptibility data are limited, especially for new antibiotics. Ceftolozane/tazobactam (C/T) is reported to be a new potent antibiotic against various pathogens. Thus, we aim to investigate C/T in vitro activity against clinical isolates from cIAI patients. This prospective cross-sectional study was conducted in three major referral hospitals in Indonesia, including Dr. Cipto Mangunkusumo Hospital (Jakarta), Dr. Kariadi Hospital (Semarang), and Dr. Soetomo Hospital (Surabaya), enrolling those diagnosed with cIAIs. Blood specimens were collected before or after at least 72 h of the last antibiotic administration. Meanwhile, tissue biopsy/aspirate specimens were collected intraoperatively. These specimens were cultured, followed by a susceptibility test for specific pathogens. The minimum inhibitory concentration (MIC) of isolates was determined according to CLSI M100. Two-hundred-and-eighty-four patients were enrolled from 2019–2021. Blood culture was dominated by Gram-positive bacteria (GPB, n = 25, 52.1%), whereas abdominal tissue culture was dominated by Gram-negative bacteria (GNB, n = 268, 79.5%). The three most common organisms were GNB, including *E. coli*, *K. pneumoniae*, and *P. aeruginosa*. C/T was susceptible in 96.7%, 70.2%, and 94.1% of the *E. coli*, *K. pneumoniae*, and *P. aeruginosa* isolates, respectively. In addition, C/T also remained active against ESBL *Enterobacterales* and carbapenem-non-susceptible *P. aeruginosa*. Overall, C/T demonstrates a high potency against GNB isolates and can be considered an agent for carbapenem-sparing strategy for cIAI patients as the susceptibility is proven.

## 1. Introduction

Infection greatly contributes to disease burden, especially in developing countries. In Indonesia, infection remains one of the leading causes of death, with respiratory and enteric infections being the utmost contributors [1]. Infections caused by bacteria resistant to one or more antibiotics have become a global challenge and have raised concern as antibiotics are losing effectiveness at a higher rate than their replacements [2]. Problems related to antimicrobial resistance (AMR) raised the need for physicians to understand the local antimicrobial susceptibility patterns to deliver appropriate management for complicated infections [3].

Intra-abdominal infections (IAIs) are tissue-invasive and occur within the abdominal cavity. Complicated intra-abdominal infections (cIAIs)—denoting visceral organ perforation—lead to significant morbidity and mortality, particularly if improperly managed. Multi-drug-resistant strains of *Enterobacterales* and *Pseudomonas aeruginosa* have been major contributors in cIAI cases [4]. These pathogens are commonly resistant due to the production of ESBL (extended-spectrum β-lactamase) enzymes. It leads the resistance to most β-lactam antimicrobial agents, including cephalosporins and monobactams, but not carbapenem. Thus, increased carbapenem use triggered carbapenem resistance growth in these organisms [5,6]. Therefore, understanding the characteristics of pathogens associated with cIAIs is important for empirical treatment decision, especially regarding appropriate antibiotic agents to prevent disease progression [7]. However, there is limited publication on microbial patterns and antibiotic susceptibility from cIAI patients in Indonesia.

Ceftolozane/tazobactam (C/T) is a combination of a novel cephalosporin and a β-lactamase inhibitor. As a β-lactam agent, ceftolozane acts by inhibiting penicillin-binding proteins (PBPs) and impairs peptidoglycan cross-linking, resulting in the degradation of the bacterial cell wall. It is known to have higher activity in *P. aeruginosa* than other β-lactams because of its stability against AmpC enzymes, active efflux, and porin channel changes. In addition, tazobactam inhibits most class A β-lactamases and some class C β-lactamases through a stable imine acyl-enzyme complex, which is hydrolyzed at a slower rate than the complex formed by other β-lactams [8,9]. This combination is believed to have high activity against a broad spectrum of *Enterobacterales*, including ESBL-producing *E. coli* and Difficult-to-Treat Resistance (DTR) *P. aeruginosa* [9].

In 2020, the Food and Drug Administration in Indonesia approved the combination of ceftolozane and tazobactam as an agent for nosocomial pneumonia, complicated urinary tract infections (cUTI), and cIAI [10]. Several studies have focused on the efficacy of C/T as a novel antibiotic regimen in cIAI. Solomkin et al., in their 2015 publication of the ASPECT-cIAI Phase 3 trial, showed that C/T plus metronidazole was non-inferior compared to meropenem in primary and secondary endpoints, meeting the pre-specified non-inferiority margin. This study also reported that in patients with ESBL-producing *Enterobacterales*, the clinical cure rates were 95.8% (23/24) and 88.5% (23/26) in the C/T plus metronidazole and meropenem groups, respectively, with identical frequency of adverse events between the two groups [11]. The susceptibility of C/T, however, has not been studied in any institution in Indonesia.

In the current study, we aimed to address the data gap mentioned above by describing the microbial patterns of cIAI patients from three referral hospitals located in Jakarta, Semarang, and Surabaya, from 2019 to 2021. We also report the activity of C/T, as a new antibiotic in Indonesia, against *Enterobacterales* and *Pseudomonas aeruginosa* compared to other broad-spectrum antibiotics. Our findings provide the first data on microbial patterns and C/T susceptibility of cIAI patients in Indonesia.

## 2. Results

### 2.1. Patient Characteristic

In the two-year study period, 284 subjects met the inclusion criteria and were enrolled in this study: 101 (35.6%) patients from Dr. Cipto Mangunkusumo Hospital (DCM), 150 (52.8%) from Dr. Kariadi Hospital (DK), and 33 (11.6%) from Dr. Soetomo Hospital (DS). Most of the patients were female (157, 55.3%), had lower intestinal tract infections (124, 43.7%), and had no comorbidity (161, 56.7%). Based on the bacterial growth in the cultures, 13.3% and 93.7% were positive in blood specimen and in abdominal tissue specimen, respectively. The detailed distribution of patients and isolates characteristics from each hospital is represented in Table 1.

### 2.2. Pathogen Distribution in Blood and Abdominal Tissue Specimens

At least 27 species of Gram-positive bacteria (GPB), 25 species of Gram-negative bacteria (GNB), and eight strains of fungus were identified in the isolates. The blood culture result was dominated by GPB (25, 52.1%), whereas GNB (268, 79.5%) was more predominant in an abdominal tissue specimen. *Staphylococcus* spp., *Streptococcus* spp., and *Enterococcus* spp. were among the utmost three GPBs isolated from the specimens. Coagulase-negative *Staphylococcus* was more common than *Staphylococcus aureus* in blood specimens and in abdominal tissue specimens (22.9% vs. 6.3% and 2.9% vs. 0.9%, respectively). On the other hand, *E. coli* was the most common GNB found, followed by *K. pneumoniae* and *P. aeruginosa*. Other species, including *A. baumannii*, *E. cloacae*, *Proteus* spp., *Citrobacter* spp., and *Prevotella* spp., were also observed in the culture. We also identified fungal growth from the culture, including *Candida* spp., *Cryptococcus* spp., and *Geotrichum* spp. The detail of each bacterium and fungus distribution (total and percentage) from all hospitals is presented in Table 2.

### 2.3. Phenotypic Subset of the Pathogen

Susceptibility testing was performed by comparing C/T and other comparators according to CLSI M100 standards. The pathogens tested for susceptibility were GNB, including, *Enterobacterales* (*E. coli*, 121 and *K. pneumonia*, 47) and *P. aeruginosa* (17) collected from blood and tissue specimens. Additionally, we grouped the *Enterobacterales* and *P. aeruginosa* isolates on their phenotypic groups based on their MIC profile.

The ESBL non-CRE phenotype was observed in 58.7% (71/121) of the *E. coli* group and 38.3% (18/47) of the *K. pneumoniae* group. In the group of *P. aeruginosa*, the phenotypic subsets isolate observed non-susceptible ceftazidime 11.8% (2/17), non-susceptible piperacillin/tazobactam 17.6% (3/17), non-susceptible meropenem 5.9% (1/17), and non-susceptible imipenem group 52.9% (9/17). The overall susceptibility of *Enterobacterales* and *P. aeruginosa* for C/T and other antibiotics is presented in Table 3.

### 2.4. Activity of C/T and Comparators against Enterobacterales and P. aeruginosa

The overall clinical isolates phenotype, antibiotics susceptibility, and MIC distribution of C/T against *Enterobacterales* isolates are presented in Table 3 and Figure 1A. Based on the current CLSI M100 guideline breakpoint, 89.3% were susceptible, 1.8% were intermediate, and 9% were resistant to C/T in *Enterobacterales* isolates. The MIC value for *Enterobacterales* ranged from ≤0.5 to ≥32 mg/L.

Further, a total of 96.7% (lowest MIC: 0.094 mg/L) of *E. coli* isolates were susceptible to C/T, which was higher than cephalosporin agents (44.6% for ceftriaxone and 52.1% for cefepime), ciprofloxacin (17.4%), ampicillin/sulbactam (41.3%), even piperacillin/tazobactam (75.2%). In addition, C/T had similar susceptibility against *E. coli* to the carbapenem group (95.2% for ertapenem, 99,2% for meropenem, and 95% for imipenem) and amikacin (95%). In contrast, only 70.2% (lowest MIC: 0.094 mg/L) of *K. pneumoniae* isolates were susceptible to C/T, which showed better in vitro activity compared to cephalosporine agents (31.9% for ceftriaxone and 34% for cefepime), ampicillin/sulbactam (26.1%), and piperacillin/tazobactam (40.4%). Also observed were the carbapenem group (75.4% for ertapenem, 80.9% for meropenem and 76.6% for imipenem) and amikacin (76.6%). Interestingly, in a phenotypic subset of ESBL non-CRE, C/T has similar activity to the carbapenem group in *E. coli* but not in *K. pneumoniae*.

Figure 1B shows the proportion of MIC distribution of C/T against *P. aeruginosa* isolates. A total of 94.1% were susceptible, and 5.9% were resistant to C/T in *P. aeruginosa* isolates (MIC range ≤ 0.25 to ≥16 mg/L, lowest MIC: 0.125 mg/L), showing superiority to other antibiotics in vitro. Ciprofloxacin (76.5%), levofloxacin (70.6%), and imipenem (47.1%) showed the lowest susceptibility in the *P. aeruginosa* group. On the other hand, C/T has varied activities in the subset group, which were 50% of CAZ-NS, 67.7% of P/T-NS, 100% of MEM-NS, and 89.9% of IMI-NS isolates.

## 3. Discussion

In this study, 284 isolates of cIAI patients from three referral hospitals showed that 266 (93.6%) growing microorganisms in culture media were predominated by GBP in blood specimens and by GNB in the abdominal specimen. Of these specimens, *Staphylococcus* spp., *Streptococcus* spp., and *Enterococcus* spp. were the three most common GPBs isolated from all specimens, whereas *E. coli* was the most common GNB, followed by *K. pneumoniae* and *P. aeruginosa.* This finding is in line with a study by Chang et al., reporting that *E. coli*, *K. pneumonia*, and *P. aeruginosa* were the three most identified GNBs from IAI patients in Asia-Pacific countries [12]. Supporting our finding, a former study in six surgical centers in Indonesia reported that GNB was the most predominant organism identified on pus culture taken from the abdominal cavity. Although *E. coli*, *K. pneumoniae*, and *E. cloacae* were mentioned as the three predominant Gram-negative pathogens, there is no *P. aeruginosa* reported between the 2015–2016 period at the DCM hospital [13]. Interestingly, our recent data showed five cases of *P. aeruginosa* (top 3 GNB) at the DCM hospital during 2019–2021, suggesting the microbial pattern shifting in cIAI over five years in the participating hospital.

The emergence of multi-drug resistance (MDR) GNB in the routinely-prescribed antibiotics with extension to newer antibiotics has been a challenge and played a critical role, especially in middle-income countries, including Indonesia [14]. However, our study’s number of ESBL phenotypes suggested a higher proportion compared to other studies in Asia-Pacific and Latin America [15,16]. In addition, our study showed a lower susceptibility of *K. pneumoniae* than other GNBs in the carbapenem group. This suggests the rise of CRE pathogens in the participating hospitals. Excessive use of carbapenem induced the production of plasmid-mediated carbapenemases responsible for CRE [17]. Another measure to overcome the problem includes choosing an antibiotic targeting a broader class of β-lactamases, such as imipenem-relebactam. Recent studies in Taiwan and the United States reported that a high proportion of CRE *E. coli* and *K. pneumoniae* were susceptible to imipenem-relebactam [18,19]. However, to date, such a combination is not yet approved in Indonesia.

Ceftolozane/tazobactam is a combination that provides potency against *P. aeruginosa* and *Enterobacterales*. To date, no study has reported C/T activity against any microbial activity in Indonesia. This study is the first to focus on the three pathogens identified in cIAI patients. Despite having a low number of isolates, our study indicates that C/T showed high activity against *P. aeruginosa* compared to other antibiotics, especially in the non-susceptible-subset. One isolate showed that C/T was the only susceptible antibiotic compared to other antibiotic groups, including carbapenem (meropenem and imipenem). This finding aligns with a previous study that reported that *Pseudomonas* non-susceptible to meropenem may still be susceptible to C/T [20,21,22]. The high susceptibility against carbapenem-resistant *P. aeruginosa* was also reported by other studies [23,24]. More local data with non-susceptible carbapenem isolates may strengthen the finding in Indonesia.

One of limitations of this study was the relatively low number of bacterial isolates, especially for *K. pneumoniae* and *P. aeruginosa.* Thus, the data may not necessarily represents the condition across Indonesia. In addition, we only assessed C/T against three GNB pathogens, despite various GNBs identified in the study, such as *Enterobacter cloacae*, *Salmonella* spp., *Proteus* spp., and others. A former study in Hongkong reported that C/T is active against both typhoidal and non-typhoidal *Salmonella*, including ESBL-producing *Salmonella* [25].

In conclusion, this study C/T showed high activity against *E. coli* and *P. aeruginosa* but lower susceptibility in *K. pneumonia*. This finding showed the importance of having and renewing local microbial patterns and antibiotic susceptibility data of cIAI patients, particularly with new antibiotics in the region. The use of C/T as an agent for carbapenem-sparing strategy in cIAI patients might be considered as the susceptibility is proven.

## 4. Materials and Methods

### 4.1. Patient Enrolment

This was a prospective cross-sectional study conducted in a laboratory. In total, 284 hospitalized patients were prospectively enrolled from three national referral hospitals in Indonesia, which included Dr. Cipto Mangunkusumo (DCM) Hospital in Jakarta, Dr. Kariadi (DK) Hospital in Semarang, and Dr. Soetomo (DS) Hospital in Surabaya from August 2019 to December 2021. Each data collection was recorded in case report form (CRF). Inclusion criteria included age category (≥18-year-old), clinical symptom, documented body temperature of ≤36.0 °C or ≥8.0 °C during the first 24 h, hospitalized patient, informed consent, and intraoperative finding(s) indicating perforated viscus. All specimens were collected from hospitalized patients who met the inclusion criteria. Patients under 18 years of age and/or diagnosed with primary peritonitis, tuberculosis peritonitis, and non-involvement of any digestive organs were excluded from this study.

### 4.2. Laboratory Procedure

Blood specimens were collected before antibiotics administration to the patient or after at least 72 h of the last antibiotic administration, whereas tissue biopsy/aspirate specimen was collected intraoperatively. Two sets of 10 mL venous blood were drawn from each patient and directly inoculated into the close system culture medium (BacTalert) for both aerobe and anaerobe examination, whereas tissue specimens were inoculated into two thioglycolate tubes for the same purposes. All the specimens were transported in less than 2 h under room temperature in the laboratory. The aerobe specimen was incubated into blood, MacConkey, and chocolate agar media, whereas the anaerobe specimen was incubated into brucella and brucella-kanamycin blood agar media. The incubation period took 18–24 h for the aerobe specimen and 24–48 for the anaerobe specimen. As a single colony developed, the identification and susceptibility test proceeded. Species identification was performed locally at each participating hospital.

Minimum inhibitory concentration (MIC) was determined for each isolate using the concentration gradient test. The antibiotic susceptibility testing (AST) profiles of isolates were then interpreted according to the CLSI M100 breakpoint [26]. For quality control (QC), *E. coli* ATCC 25922 and *P. aeruginosa* ATCC 27853 were used as the positive and negative control strain. Only *Enterobacterales* (*E. coli* and *K. pneumoniae*) and *P. aeruginosa* isolates (n = 185 isolates) were tested for antibiotic susceptibility (ETEST^®^), comprising 121 *E. coli*, 47 *K. pneumoniae*, and 17 *P. aeruginosa* isolates. Antibiotic classes tested in this study were cephalosporins (ceftriaxone, cefepime, ceftazidime), carbapenems (ertapenem, meropenem, imipenem), quinolones (ciprofloxacin, levofloxacin), aminoglycosides (amikacin), and β-lactam/β-lactamase inhibitor combinations (ceftolozane/tazobactam, ampicillin/sulbactam, piperacillin/tazobactam). Antibiotics with intrinsic resistance (IR) to *P. aeruginosa* isolates were not tested.

Moreover, VITEK-2^®^ (broth microdilution method) was used for the confirmation of ESBL producing *Enterobacterales*, and antimicrobial concentration was interpreted according to CLSI M100. Then, the ESBL non-CRE subset was determined in ESBL confirmed isolates if they were susceptible to Ertapenem (ETEST^®^, MIC < 0.5 mg/L). *P. aeruginosa* subsets were determined if the isolate were non-susceptible (NS) to ceftazidime, piperacillin/tazobactam, imipenem, and meropenem [26].

The detail of laboratory workflow for this study is described in Figure 2 below.

## Figures and Tables

**Figure 1 antibiotics-12-00052-f001:**
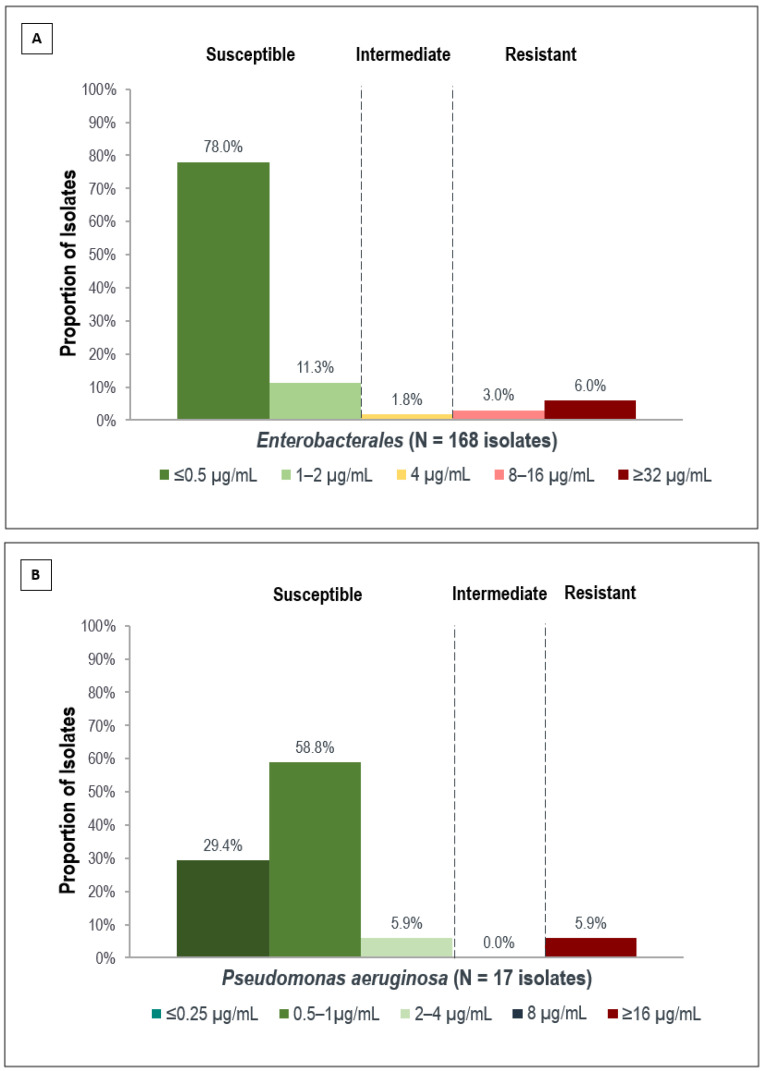
The proportion of MIC distribution of C/T against *Enterobacterales* (**A**) and *P. aeruginosa* (**B**). MIC breakpoints based on CLSI 2022 guidance. C/T MIC for *Enterobacterales* and *P. aeruginosa* are ≤2 µg/mL and ≤4 µg/mL, respectively. MIC: minimum inhibitory concentration; C/T: ceftolozane-tazobactam.

**Figure 2 antibiotics-12-00052-f002:**
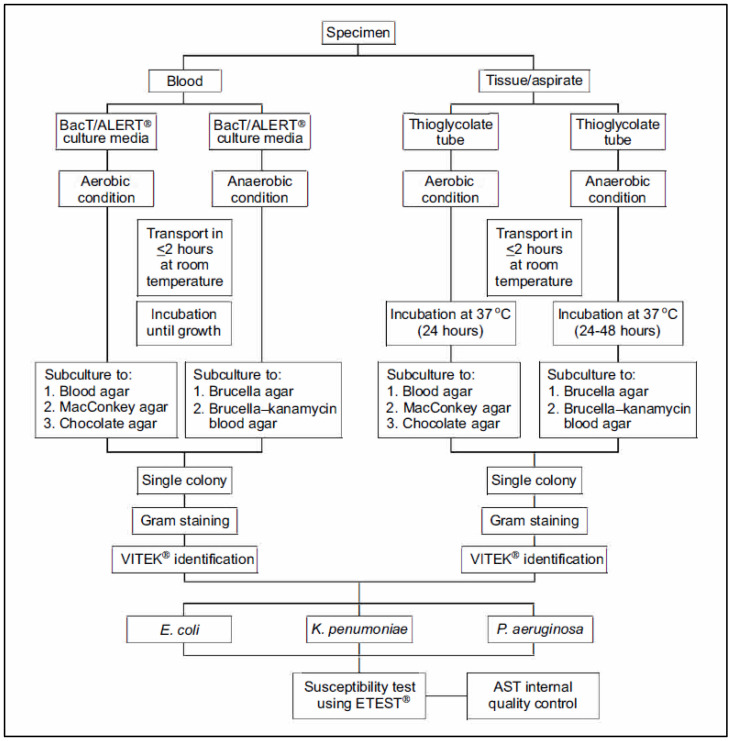
Laboratory algorithm for microbiology examination.

**Table 1 antibiotics-12-00052-t001:** Characteristics of patients and isolate growths from three referral hospitals in Indonesia from 2019–2021.

Characteristics	Number of Samples, n (%)			Total (n)
	DCM	DK	DS	
*Gender*				
Male	51 (50.5%)	64 (42.7%)	12 (36.4%)	127
Female	50 (49.5%)	86 (57.3%)	21 (63.6%)	157
*Site of infection*				
Upper intestinal tract	20 (19.8%)	33 (22.0%)	3 (9.1%)	56
Lower intestinal tract	30 (29.7%)	74 (49.3%)	20 (60.6%)	124
Other abdominal cavity organs	51 (50.5%)	36 (24.0%)	8 (24.2%)	95
Unknown	0 (0.0%)	7 (4.7%)	2 (6.1%)	9
*Comorbidity status*				
With comorbidity	47 (46.5%)	70 (46.7%)	1 (3.0%)	118
Without comorbidity	54 (53.4%)	75 (50.0%)	32 (97.0%)	161
Unknown	0 (0.0%)	5 (3.3%)	0 (0.0%)	5
*Bacterial Growth (Blood)*				
Mono-bacterial (1 species)	19 (18.8%)	4 (2.7%)	5 (15.2%)	28
Poli-bacterial (>2 species)	9 (8.9%)	0 (0.0%)	0 (0.0%)	9
No growth	73 (72.3%)	143 (95.3%)	28 (84.8%)	244
Unknown	0 (0.0%)	3 (2.0%)	0 (0.0%)	3
*Bacterial Growth (Abdominal Tissue)*				
Mono-bacterial (1 species)	42 (41.6%)	145 (96.7%)	26 (78.8%)	213
Poli-bacterial (>2 species)	47 (46.5%)	1 (0.7%)	5 (12.1%)	53
No growth	12 (11.9%)	1 (0.7%)	2 (6.1%)	15
Unknown	0 (0.0%)	3 (2.0%)	0 (0.0%)	3

DCM: Dr. Cipto Mangunkusumo Hospital (Jakarta); DK: Dr. Karadi Hospital (Semarang); DS: Dr. Soetomo Hospital (Surabaya).

**Table 2 antibiotics-12-00052-t002:** Identified pathogens from blood and abdominal tissue specimens from 2019–2021.

Microbes	Number of Isolates			Total n (%)
	DCM	DK	DS	
** *Blood Specimen* **				
**Gram-positive bacteria**	**22**	**1**	**2**	**25 (52.1)**
*Enterococcus faecalis*	3	-	-	3 (6.3)
*Streptococcus agalactiae*	1	-	-	1 (0.9)
*Streptococcus suis*	1			1 (0.9)
*Staphylococcus aureus*	1	1	1	3 (6.3)
Coagulase-negative *Staphylococcus*	11	-	-	11 (22.9)
Other species	5	-	1	6 (12.5)
**Gram-negative bacteria**	**11**	**3**	**3**	**17 (35.4)**
*Acinetobacter baumanii*	2	-	-	2 (4.2)
*Klebsiella pneumoniae*	2		1	3 (6.3)
*Escherichia coli*	4	3	2	9 (18.7)
*Pseudomonas aeruginosa*	1	-	-	1 (2.1)
Other species	2	-	-	2 (4.1)
**Fungi**	**6**	**0**	**0**	**6 (12.5)**
*Candida* spp.	5	-	-	5 (10.4)
*Cryptococcus laurentii*	1	-	-	1 (2.1)
*Abdominal Tissue Specimen*				
**Gram-positive bacteria**	**46**	**10**	**0**	**56 (16.6)**
*Enterococcus* spp.	29	3	-	32 (9.5)
*Staphylococcus aureus*	1	2	-	3 (0.9)
Coagulase negative-*Staphylococcus*	5	5	-	10 (2.9)
*Streptococcus viridians group.*	8	-	-	8 (2.4)
Other species	3	-	-	3 (0.9)
**Gram-negative bacteria**	**99**	**137**	**32**	**268 (79.5)**
*Acinetobacter baumannii*	1	1	-	2 (0.6)
*Enterobacter cloacae*	3	2	3	8 (2.4)
*Escherichia coli*	55	86	21	162 (48.1)
*Klebsiella pneumoniae*	22	20	5	47 (13.9)
*Klebsiella* spp.	-	2	2	4 (1.2)
*Pseudomonas aeruginosa*	5	16	-	21 (6.2)
*Proteus* spp.	2	5	1	8 (2.4)
*Citrobacter* spp.	2	1	-	3 (0.9)
*Prevotella* spp.	4	-	-	4 (1.2)
Other species	5	4	-	9 (2.6)
**Fungi**	**12**	**1**	**0**	**13 (3.9)**
*Candida* spp.	7	1	-	8 (2.4)
*Geotrichum klebahnii*	1	-	-	1 (0.3)
*Cryptococcus laurentii*	4	-	-	4 (1.2)

DCM: Dr. Cipto Mangunkusumo Hospital (Jakarta); DK: Dr. Karadi Hospital (Semarang); DS: Dr. Soetomo Hospital (Surabaya).

**Table 3 antibiotics-12-00052-t003:** Phenotypic subsets and level of antibiotics susceptibility.

Pathogen	N	% of Susceptibility											
		C/T	P/T	ASB	ERT	IMI	MEM	FEP	CAZ	CRO	AMK	LEV	CIP
*E. coli*	121	96.7	75.2	41.3	97.5	95	99.2	52.1	NA	44.6	95	NA	17.4
ESBL non-CRE *	71	94.4	69	29.6	-	94.4	98.6	23.9	NA	14.1	91.5	NA	9.9
*K. pneumoniae*	47	70.2	40.4	26.1	74.5	76.6	80.9	34	NA	31.9	76.6	NA	31.9
ESBL non-CRE *	18	77.8	38.9	22.2	-	94.4	100	22.2	NA	16.7	94.4	NA	11.1
*P. aeruginosa*	17	94.1	82.4	NA	NA	47.1	88.2	82.4	88.2	NA	88.2	70.6	76.5
CAZ-NS	2	50	0	NA	NA	0	0	0	-	NA	0	0	0
P/T-NS	3	67.7	-	NA	NA	0	33.3	0	33.3	NA	33.3	33.3	33.3
MEM-NS	1	100	0	NA	NA	0	-	0	0	NA	0	0	0
IMI-NS	9	89.9	66.7	NA	NA	-	77.8	66.7	77.8	NA	77.8	77.8	77.8

C/T: ceftolozane/tazobactam; P/T: piperacillin/tazobactam; ASB: ampicillin/sulbactam; ERT: ertapenem; MEM: meropenem; IMI: imipenem; AMK: amikacin; CIP: ciprofloxacin; LEV: levofloxacin; CRO: ceftriaxone; FEP: cefepime; CAZ: ceftazidime; ESBL: extended spectrum β-lactamase; CRE: carbapenem-resistant *Enterobacterales*; NS: non-susceptible, NA: not available/tested. * ESBL confirmed by VITEK-2^®^ and ertapenem-susceptible (MIC < 0.5 mg/L) were used as phenotype indicator.

## Data Availability

Data available on request due to privacy or ethical restrictions.

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
