# Peer review of "Ceftolozane/Tazobactam In-Vitro Activity against Clinical Isolates from Complicated Intra-Abdominal Infection Patients in Three Indonesian Referral Hospitals"

_antibiotics, 2022, doi:10.3390/antibiotics12010052_

Round 1
Reviewer 1 Report
The authors described the antimicrobial susceptibility of 185 isolates (distributed among E. coli, K. pneumoniae and P. aeruginosa) recovered in 3 hospitals in Indonesia.
The major issue to be clarify is how the authors inferred the presence of ESBL, which is mentioned in results section and discussed in the discussion section. They didn’t explain ESBL detection in materials and methods section. Did the use the Vitek screening?
If they desire to discuss about ESBL production among the studied isolates, I recommend performing the ESBL test described in table 3A of CLSI 2022 in E. coli and K. pneumoniae isolates. In P. aeruginosa double disk synergy method using CTX and CAZ or cefepime (because of AmpC) disks in combination with clavulanic acid could be assayed.
I suggest focus the objective on the microbial pattern of gram-negative bacteria and remove gram positive from the Table 2 (even you can mention in the text). The data of the complete microorganisms that were recovered can be included (gram positive + gram negative + fungi) as supplementary data.
Taking into account the Magiorakos et al publication (DOI: 10.1111/j.1469-0691.2011.03570.x), can the authors categorized the isolates into MDR, XDR or PDR? After that, the evolution of their profile (considering the period 2019 through 2021 which contains COVID-19 Pandemic) can be analyzed.
Describe in Materials and Method the period of analysis.
How many beds have each hospital? The proportion of the specimens is related with the size of each hospital?
Table 3. Order the antibiotics by family and generation inside them. It’s difficult to analyze as they presented the results.
English could be improved.
Minor comments.
Line 36. The authors said….”C/T was susceptible”….and should be…..”96.7%, 70.2%, and 94.1% of the E. coli, K. pneumoniae, and P. aeruginosa 36 isolates, respectively were susceptible to C/T.
Line 103. Remove below after Table 1.
Line 132. The author said that ESBL non-CRE phenotype was observed. In materials and method, they didn’t describe how the inferred the presence of an ESBL. Did the use the screened performed by VITEK? Explain with more detail.
Line 134. What the meaning of the acronymous NS after ceftazidime, piperacillin/tazobactam, meropenem, imipenem. Is NS not susceptible?
Figure 1. Change the reference color. Put resistant in dark red and susceptible in green.
Line 166 and 183. What is PsA group? P. aeruginosa group?
Line 183-184. You didn’t discriminate the recovery by year in any table. How did you asses this?
Line 170. You said, ……”In this study, we described the microbial pattern and antibiotic susceptibility of 284 cIAI patients from 3 different centers”……But in reality, you described the susceptibility of the bacterial isolates not of the patients (which don’t exists). Re-write the sentence.
Line 232. Change by was screened.
Line 258-260. Remove the sentence. It is not necessary definition of Etest.
Line 264. You should use P. aeruginosa isolates.
Line 265-266. Not necessary to explain, it is the control of the method. If you desire, only put E .coli ATCC 25922 & P. aeruginosa ATCC 27853 were used as control strains.
Author Response
Response to Reviewer 1
Dear Reviewer,
Thank you very much for your comment on our manuscript “Ceftolozane/Tazobactam In-vitro Activity Against Clinical Isolates from Complicated Intra-abdominal Infection Patients in Three Indonesian Referral Hospitals”.
Here our response for some of the comments provided:
- ESBL detection: We were utilizing VITEK-2 to confirm the presence of ESBL for our research model (Interpreted based on CLSI criteria). VITEK-2 should be sufficient and additional test with double-disk method don’t need to be performed further. To explain the phenotypic subset, we adding a paragraph in the method section (Line 259-263).
- Categorizing drug resistant & COVID Impact analysis: This is an interesting approach for the data. However, as this is a descriptive study, we did not aim to compared the data Pre- & During COVID-19 Pandemic, but describing microbial pattern and susceptibility. Additionally, the data will not be sufficient enough to run some statistical analysis due to lack of patient enrolment in one centre.
- Hospital situation: Initially, we tried to collect 100 isolates from each centre. But, due to high cases of COVID in Indonesia, one of our centres (Soetomo Hospital, Surabaya) was not able to enrol any patient that meet the inclusion criteria. Therefore, the proportion of patients are not evenly distributed as planned.
- Order of antibiotic family: Edit as suggested.
- Minor comments: All edited according to the comment.
Please find the edited manuscript attached. We hope the revision is enough to answer all your comments.
Regards,
Amril on behalf of All Authors

Reviewer 2 Report
A scientifically sound manuscript that needs a few minor corrections before being published. See below:
* Double plural (...Infections Patients...) in title needs correcting
* English language review needed
* 'Gram' must be capitalised in all instances throughout manuscript
* Names of bacterial species should always be italicised (please check all instances)
* Latin phrases should be italicised in all cases (e.g. in vitro) (please check all instances of all latin phrases used)
* Drug names should not be capitalised unless first in a sentence
* With numbers/percentages, sometimes the authors use a comma instead of a decimal point. Please correct this throughout (should be decimal point).
* The legend for Table 1 is split by the table itself (it appears both above and below the table). Please make it entirely above or entirely below the table.
* In lists of bacterial species names, connecting words like 'and' are sometimes italicised too. Please correct this mistake throughout manuscript.
* Sometimes the word 'table' is capitalised ('Table 1') and sometimes not ('table 3'). Please be consistent.
* Please check all references thoroughly for typos (e.g. date incorrect in Ref #25). Also lots of missing spaces in many references.
Author Response
Dear Reviewer 2,
Please kindly find attached the revision based on your comment.
We did re-proofread and edit it to improve the english writing for the manuscript.
All reference have been checked and revised.
Best Regards,
Amril on behalf of other authors.

Round 2
Reviewer 1 Report
I received the revised version which is very improved. I would like to ask the following few minor corrections to the authors.
Table 3. The antibiotics are not in order. Considering more powerful drug first, (as they listed) the order should be:
C/T, P/T, ASB, ERT, IMI, MEM, FEP, CAZ, CRO, AMK, LEV, CIP
Or even you can use:
ASB, P/T, C/T, CAZ, CRO, FEP, ERT, IMI, MEM, AMK, CIP, LEV.
Line 177: correct K. pneumonia and E. cloaca, lack an e at the end.
In material and methods, line 259. Vitek-2 in not a confirmatory test for ESBL. In accordance with CLSI, the phenotypic methods that can be mentioned as confirmatory test for ESBL presence are those listed in Table 3A from CLSI, or you can confirm by genotypic methods listed in appendix H from CLSI.
The authors decide, you can reformulate that sentence saying that you only screened ESBL by VITEK-2 without any test for confirmation. But if you decide to say that you CONFIRMED the ESBL presence you must perform a confirmatory test for ESBL.
The name of the author Kuntaman seems to be incomplete.
Author Response
Dear Reviewer,
Thanks for your comment. We edited some minor comment accordingly.
For ESBL confirmation, we believe that VITEK should be sufficient as confirmatory not only for screening. As written in the insert package, VITEK-2 utilized broth microdilution method and the machine interpret ESBL by concentration according to CSLI M100 Table 3A (CTX + CTX/CA & CAZ + CAZ/CA).
Henceforth, VITEK is just the machine, but we use the method and interpretation based om CLSI. This should be enough to support the presence of ESBL in our isolates and double-disk method is not needed.
Warm Regards,
Amril